# Action Mechanisms and Scientific Rationale of Using Nasal Vaccine (HeberNasvac) for the Treatment of Chronic Hepatitis B

**DOI:** 10.3390/vaccines10122087

**Published:** 2022-12-07

**Authors:** Julio Cesar Aguilar, Jorge Agustin Aguiar, Sheikh Mohammad Fazle Akbar

**Affiliations:** 1Hepatitis B Therapeutic Vaccine Project, Center for Genetic Engineering and Biotechnology, La Habana 10600, Cuba; 2Department of Gastroenterology and Metabology, Ehime University Graduate School of Medicine, Matsuyama 791-0295, Japan

**Keywords:** HeberNasvac, chronic hepatitis B, therapeutic vaccine, innate immunity, acute respiratory infections

## Abstract

Nasvac (HeberNasvac^®^) is a novel therapeutic vaccine for chronic hepatitis B (CHB). This product is a formulation of the core (HBcAg) and surface (HBsAg) antigens of the hepatitis B virus (HBV), administered by nasal and subcutaneous routes, in a distinctive schedule of immunizations. In the present review article, we discuss the action mechanisms of HeberNasvac, considering the immunological properties of the product and their antigens. Specifically, we discuss the capacity of HBcAg to activate different pathways of innate immunity and the signal transduction after a multi-TLR agonist effect, and we review the results of recent clinical trials and in vitro studies. Aimed at understanding the clinical results of Nasvac and other therapeutic vaccines under development, we discuss the rationale of administering a therapeutic vaccine through the nasal route and also the current alternatives to combine therapeutic vaccines and antivirals (NUCs). We also disclose potential applications of this product in novel fields of immunotherapy.

## 1. Introduction

Hepatitis B virus (HBV) is a non-cytopathic, hepatotropic virus, able to induce a persistent infection, which might lead to acute and chronic diseases. The HBV genome consists of a small DNA genome, which is a partially double-stranded and relaxed circular DNA (rcDNA) molecule. The HBV attacks the liver and can cause both acute and chronic disease. The WHO estimates that 296 million people were living with chronic hepatitis B (CHB) infection in 2019, with 1.5 million new infections each year. In 2019, hepatitis B resulted in an estimated 820,000 deaths, mostly from cirrhosis and hepatocellular carcinoma [1], 30.4 million people (10.5% of all people estimated to be living with hepatitis B) were aware of their infection, and 6.6 million (22%) were on treatment [1].

Two types of products have been approved for CHB treatment: Peginterferon (PEG-IFN), with antiviral and immune modulatory properties and nucleoside/nucleotide analogs (NUCs), direct inhibitors of the reverse transcriptase enzyme interfering in viral replication. These drugs have poor efficacy in terms of sustained post-treatment viral suppression and generate important secondary effects during and after therapy. Although these products have been studied in combination, at present, the coadministration is not recommended [2]. 

The treatment of CHB requires the commitment of patients and a solid healthcare system. Apart from product expenses and limitations, additional costs are required to study HBV DNA levels during 3 to 6 months, as well as biochemical, serological, and histological characterizations of patients are needed for treatment decision. Periodical evaluation on therapy further complicates the follow-up of patients, especially in the case of PEG-IFN due to multiple adverse reactions [2]. Consequently, inexpensive, non-reactogenic, finite, and liver-protecting therapies are desirable qualities for novel and competitive products. Therapeutic vaccines are easy to administer and have the potential to be a finite and a first-line treatment even in those that are not covered by current treatment recommendations. However, the introduction of a therapeutic vaccine in the field of CHB treatment has proven challenging.

## 2. Nasvac: A Novel Vaccine for Chronic Hepatitis B Treatment

Nasvac is a novel mucosal and parenteral therapeutic vaccine designed and developed for the treatment of CHB. This is a liquid formulation containing the recombinant hepatitis B core and surface antigens (HBcAg and HBsAg, respectively), and it is produced at the Center for Genetic Engineering and Biotechnology (CIGB) in Havana, Cuba [3]. The formulation contains 100 μg of each virus-like particle antigen in 1.0 mL of phosphate-buffered saline (PBS). Due to its stability, this product does not require preservatives or stabilizers. The proteoliposomal and nucleoprotein composition of HBsAg and HBcAg, with a mean size of 22 and 32 nm, respectively, and a virus-like particle physical aspect, have a remarkable immunomodulatory effect, stimulating innate immunity and adaptive immune effector cells [4,5]. Nasvac has completed preclinical and clinical steps in Cuba and Bangladesh as part of an international effort with scientists from Cuba, Bangladesh, Japan, Australia, France, and Germany contributing [3,4,5,6,7,8].

The regulatory process is advancing at different speeds in several countries, where the product has been studied as a monotherapy and also as a combined treatment with NUCs [9,10]. At present, it has been registered for monotherapy of CHB in Cuba [3], and different studies are optimizing the use of this product as a combined therapy; however, the combination of immunomodulatory drugs and antivirals deserves immunological understanding [2,10,11].

The major pharmacological findings related to the action mechanism and the scientific rationale of the nasally administered vaccine are compiled. As the ultimate goal of CHB treatment is to prevent the long-term progression to end-stage liver disease, the effect of HeberNasvac on liver disease progression will be revised, as well as the future prospects related to this novel product.

## 3. Rationale of Using the Nasal Route of Immunization

### 3.1. Association of the i.n. Immunization to Immune Activation and HBV Clearance

The phase 3 trial conducted in Bangladesh (NCT01374308) compared Nasvac to PEG-IFN treatment in an open, randomized, controlled, two-arm trial. The vaccine was administered to 160 CHB patients in two cycles first, with inoculations at Weeks 0, 2, 4, 6, and 8 by the i.n. route only and, second, at Weeks 12, 14, 16, 18, and 20, administering the product by both the i.n. and s.c. routes, simultaneously [8]. The reduced number and frequency of adverse events detected in Nasvac-treated patients compared to the PEG-IFN group demonstrated its safety. Transient and generalized ALT increases, above twice the upper limit of normal, were detected in the Nasvac group by Week 12. This effect, found after the first cycle of five immunizations, was considered a benign consequence of the immune response induced in the liver. In contrast, this effect was detected in a few patients who received PEG-IFN. Nasvac induced ALT increases regardless of patient HBeAg status, sex, age, or initial viral load and were not related to signs of decompensation [8]. The PEG-IFN-induced ALT increases had a similar range of intensity, but occurred at different time points during the 48 weeks of treatment. A generalized control of HBV DNA detected after the i.n. immunization cycle evidenced that vaccine-induced immune activation was linked to the recognition and elimination of infected hepatocytes, although the limited levels of ALT increases suggest that cytolysis was not the major mechanism of viral clearance. In summary, the i.n. route of administration was a relevant factor of vaccine efficacy and part of the action mechanism of Nasvac considering the non-cytopathic nature of the HBV lifecycle and the temporary association to the end of the i.n. cycle [8].

### 3.2. Unique Properties of the Nasal Route in Animal Models of HBV Immunotolerance

Going back to the bench, a group of investigators working at Pasteur Institute revealed the impact of the nasal route of administration on the immune and antiviral responses in naïve and hepatitis-B-virus (HBV)-carrier mouse models [12]. This study evidenced that the Nasvac therapeutic vaccine elicits immune responses, and the nasal route was able to overcome tolerance and elicit a unique and polyfunctional CD4^+^ T-cell response. The i.n. route was the most efficacious at inducing cellular immune responses, in particular CD4^+^ T-cells. Cellular responses in the spleen were strong and mainly due to CD4^+^ T-cells. High frequencies of HB-specific CD4^+^ T-cells secreting interferon (IFN)-γ, interleukin (IL)-2 and tumor necrosis factor (TNF)-α were found in the liver only after i.n. immunization (Figure 1).

Increased frequencies of CD4^+^ T-cells expressing the integrin CD49a in the liver suggest a role of the nasal route in the cellular homing process [12]. Multiple dose schedules of immunizations also appear to be a prerequisite for protein-based immunization in order to overcome immunotolerance in HBV-carrier mice [13,14]. These findings provide a new rationale for further preclinical and clinical development of an i.n.-only vaccine formulation by optimizing the route and schedule of immunizations (Table 1). 

### 3.3. Long-Term Follow-Up of CHB Patients Treated with Nasvac by Nasal Route Only

A long-term follow-up of the Nasvac phase I clinical trial was conducted to explore the long-term effect of i.n. Nasvac in a group of IFN refractory patients that were treated with 10 (i.n.-only) administrations every 14 days [17]. These patients had documented histories of over 12 years with CHB that had been refractory to IFN-alpha 10 years before vaccination. Other complications such as history of cancer, alcohol consumption, and advanced age appeared in three cases. Interestingly, after five years follow-up, HBsAg became undetectable in 2 out of the 6 patients and seroconverted to anti-HBs sustainedly at the end of the follow-up term. All three HBeAg-positive patients became negative and two became anti-HBeAg-positive [17]. ALT elevations were transient and temporarily associated with HBeAg elimination and HBV DNA reduction in the three HBeAg-positive patients, suggestive of immune-mediated antiviral activity. All patients ended the 5 years follow-up with low fibroscan levels. The delayed ALT increases may be explained in this case as a result of a late effect of the product activation in this setting of patients, refractory to interferon alpha 2b treatment, with an older age and, consequently, long-term HBV carriers.

### 3.4. Nasal Administration of Nasvac in the Setting of Inactive Carriers

A nasal study conducted at Ehime Hospital, Japan, used a formulation of Nasvac and the mucoadhesive CVP (Toko Yakuhin, Kogyo, Japan), formulated before administration. HBV carriers were immunized by the i.n. route for 10 i.n. doses every 14 days. The recently published paper reported 11.1% (4 out of 36 patients) with HBsAg elimination after 18 months follow-up, while at 6 months follow-up, only 2 out of 36 patients (5.6%) were negative for HBsAg [9]. This initial assessment of the long-term effect was consistent with the previously detected late effect after 5 years follow-up and with the role of intranasal administration [17]. 

## 4. On the Rationale of Combining NUCs and Therapeutic Vaccination

### 4.1. Vaccination under NUCs: Pros and Cons

The combination of PEG-IFN and NUCs, once considered as a potential royal wedding, has not yet been proven to have a sufficient effect to become a recommendation for CHB treatment. Similarly, the failure of therapeutic vaccines—also a product of the immunomodulatory nature like PEG-IFN—should be analyzed considering that most relevant trials have been conducted in patients under a deep and long-standing viral suppression [18,19,20]. 

The combination of therapeutic vaccines and NUCs was justified on the observation that the decrease in the HBV viral load often precedes the detection of anti-HBV-specific T-cell responses, both in patients resolving natural infections and in those achieving HBeAg seroconversion. In addition, a reduction in HBV load using NUCs increases the responsiveness of HBV-specific T-cells, which are hypo-responsive in cases of persistent HBV infection [21]. However, the HBV-specific T-cell responses are only detectable during the first few months of lamivudine treatment [22], and this restoration of T-cell activity is partial, transient, and does not lead to an increase in HBeAg seroconversion [23]. 

In general, clinical trials of therapeutic vaccination under NUCs, followed by treatment discontinuation, have been designed to restart NUCs treatment just after the first detection of HBV DNA or when there was evidence of HBV replication at low levels [18,19,20]. This clinical design has been criticized as a very conservative approach that prevents the potential immune reactivation, DNA control post rebound, and HBsAg elimination in a significant proportion of patients [24]. 

### 4.2. Nasvac Vaccination and NUCs Cessation

The clinical development of Nasvac has followed different approaches in its unavoidable interaction with antivirals. The first clinical evaluation of HeberNasvac combined with NUCs failed to demonstrate the expected efficacy ([10], revised in [25]). Nasvac was studied in a phase IIb trial conducted in Asian countries (NCT02249988). The study assessed the same vaccine formulation (Nasvac) in terms of quality specifications, dose, and administration schedule (described in Section 3.1). In this study, the vaccine was administered to virally suppressed patients. The antiviral treatment continued up to one month after the end of vaccinations. The study assessed Nasvac vaccination in HBeAg(-) CHB patients under antiviral treatment for several years, evaluating the capacity of this treatment to prevent relapse after stopping antiviral therapy with NUCs. A total of 276 HBeAg(-) non-cirrhotic patients, who had been treated for at least 2 years with NUCs and who were HBV DNA-negative with normal ALT levels, were enrolled. This treatment was compared against NUCs only (n = 92). At Week 24, antiviral therapy was stopped in all patients. The patients were followed for 24 weeks, being reinserted in antiviral treatment if reaching 10,000 copies/mL. The primary endpoint of the study was the percentage of subjects who maintained HBV DNA levels 1000 IU/mL. Nasvac vaccination was safe and well tolerated, with only 2.2% severe adverse events in both treatment arms (not drug related). This HBV therapeutic vaccine study, which was also the largest prospective study stopping NUCs so far, showed that the vaccination was safe; however, it did not prevent viral relapse after stopping NUCs. Furthermore, it revealed an unexpected relapse timing difference between TDF and ETV ([10], revised in [25]).

### 4.3. Immunotherapy under NUCs: Immunological Perspective

The therapeutic vaccination of patients under long-term antiviral treatment should consider potential limitations that more rational approaches may circumvent. Vaccine-induced T-cells should exert their function in the liver. However, due to the treatment with NUCs, the anti-inflammatory liver environment is reinforced, as reflected by the reduction in alanine amino transferase (ALT) levels in most patients a few weeks after treatment starts [19,20,26,27,28]. In line with this, it has been demonstrated that hepatocytes do not express HLA class II, except under inflammatory conditions [29,30,31], a logical and natural adaptation to the tolerogenic role of the liver, further reducing the antigenicity of hepatocytes in patients without treatment.

The low viral replication in patients under NUCs therapy for several years leads to a reduction in the number of hepatocytes presenting viral antigens, for example cytoplasmic HBcAg. It has been demonstrated that the control of the replication can be predicted by the low intracellular expression of HBcAg [32]. Taken together, in virally suppressed patients, there is a logically reduced presentation of HBV antigens and, consequently, a reduction in the presentation of viral peptides to vaccine-induced T-cells, both by HLA class I and II molecules. Viral replication after NUCs cessation reactivates the immune system detection mechanisms, restarting the immune response. The discontinuation of NUCs treatment has a potential adjuvant effect on vaccine-induced immune response; however, the vaccination discontinuation protocol needs to be tailored to optimize the immune response function after the reappearance of the antigen. When to stop and restart NUCs becomes part of the action mechanism for the therapeutic vaccine.

### 4.4. NUCs Stopping Rules: Learning from Nature

The HBsAg loss during NUCs treatment is a rare event [33]. In addition, most CHB patients receive NUCs therapy for several years. Three stopping rules have been defined for discontinuation of NUCs: (1) if HBsAg is negative with or without anti-HBs antibodies; (2) in patients with HBeAg-positive CHB NUCs treatment can be discontinued if they achieve stable HBeAg seroconversion, HBV DNA undetectability, and 6 to 12 months of consolidation therapy; (3) in HBeAg-negative patients, discontinuation of NUCs was considered as a recommendation for the first time in European guidelines for non-cirrhotic HBeAg-negative patients who have achieved HBV suppression under NUCs during three years, provided that a close post-treatment monitoring of DNA and ALT can be guaranteed (reviewed in [25]). 

A review and metanalysis [24] compiled the clinical experiences of treatment discontinuation in support of this novel recommendation, evidencing the benefit of treatment cessation for an important proportion of HBeAg-negative patients [34,35,36]. The EASL 2017 Clinical Practice Guidelines (CPGs) panel considered that maintenance of the inactive carrier state seems to be the most clinically relevant endpoint for patients discontinuing NUCs. They considered that HBV DNA levels below 2000 UI/mL accompanied by normal ALT activity, instead of undetectable HBV DNA, represent a reasonable definition of post-therapy remission. However, according to the Panel, the conservative definition of post-NUCs remission may lead to unnecessary early retreatment. Upon cessation of therapy, HBeAg-negative patients often develop viral rebound and early transient “beneficial flares” associated with the mounting of an adaptive immune response. This may lead to long-lasting remission and spontaneous HBsAg clearance in an important proportion of these patients [24]. The early retreatment with NUCs would suppress this natural therapeutic benefit after discontinuation.

### 4.5. Therapeutic Avenues Using Nasvac in the Setting of Patients with NUCs

The clinicians may choose to continue a long-lasting therapy with NUCs for multiple reasons. One interesting avenue has been to administer Nasvac without stopping NUCs. In this setting, Nasvac formulated with a mucoadhesive was administered to CHB patients by the i.n. route alone. The hope was that vaccination would reduce the long-lasting period of time required to eliminate surface antigens (from 0.3% to 1% annual rate of HBsAg loss). Preliminary results after 18 months follow-up showed 7.4% HBsAg loss when Nasvac was administered by the i.n. route in a formulation with a novel mucoadhesive in a 10-dose schedule every 2 weeks [9]. The acceleration of HBsAg reduction has been another attractive result in this trial. These results need confirmation in a large and controlled clinical trial. Patients should be followed for a longer period of time and eventually boosted in case the trend of HBsAg control is reduced.

A second approach will study Nasvac applied to patients just after NUCs cessation. This study will bring together the natural effect of immune activation after natural HBV DNA rebound together with the induction of a potent immune response after vaccination. In this study, the patients receive the vaccine just after the cessation of NUCs in a ten-dose schedule every 2 weeks. The virological and serological variables will be compared with those from patients that stop NUCs or remain under antiviral treatment. The study is aimed to study the safety and efficacy variables for 3, 6, and 12 months after NUCs cessation. The authors are looking for a safer and more effective way to discontinue antiviral treatment.

## 5. On the Immunomodulatory Properties of Nasvac Antigens

The HBV antigens used in Nasvac, HBsAg and HBcAg, are virus-like particles of proteoliposomal and nucleoprotein composition, respectively. Different studies in animal models and in vitro have shown their capacity to induce a strong stimulation of dendritic cells [4] and the activation of B- and T-cells [5], consistent with the development of a strong adaptive Th1 immune response [37]. At the molecular level, HBcAg showed a multi-Toll-like receptor (multi-TLR) agonist effect, stimulating human hepatoma (HepaRG) cells (infected or not with the HBV), increasing the expression of Toll-Like Receptor 2 (TLR2), TLR3, TLR7, TLR8, and TLR9 and their signal transduction pathways: Myeloid Differentiation Factor 88 (MyD88), TRIF, IRF3, and NF-kappa B (NF-kB) [15,16]. 

The high RNA content of the HBcAg, between 15% and 25% of RNA/protein (*w*:*w*), justifies the capacity of this compound to stimulate multiple TLRs, activating both the MyD88-dependent and TRIF-dependent pathways. This combined stimulation translates into a synergistic effect [38,39], which increases the induction of class I/II HLA co-stimulatory molecules B7.1 and B7.2 (CD80/86), type I interferons, and other cytokines, a set of molecules with a fundamental role in antiviral immune defense [15,16]. The agonistic effect of HBcAg on TLR2 and TLR7 has already been described [38,39]; however, the HBcAg used in Nasvac is unique in its capacity to stimulate TRIF- and MyD88-dependent and synergistic pathways [15,16], a remarkable and useful quality for a single compound that opens the door to several applications in the field of infectious diseases and oncology.

The multi-TLR agonist effect is relevant in CHB immunotherapy considering the strategies of HBV to evade the innate immune response and induce immunosuppression. PBMCs isolated from CHB patients have decreased expression of TRIF, attenuating TLR3 and TLR4 signaling subsequent to HBV infection. This has been considered as one of the reasons why HBV infection is stable in CHB patients, as recently reviewed by Xu and colleagues [40]. TLR2 is involved in the recognition of HBV during the early infection process and initiates innate immune signaling in primary human hepatocytes. Several TLRs, including TLR3, TLR4, TLR7, and TLR9, are downregulated in CHB patients. In vitro studies using HBcAg evidenced a strong reduction in HBV replication in cultured HepaRG cells [15]. This pharmacological effect of HBcAg on the innate immune stimulation—as an IFN inducer—may also mediate a potential antiviral effect in addition to the adaptive immune response in vivo.

The HBcAg-containing formulations have been used in the field of post-exposure prophylaxis of SARS-CoV-2 and respiratory infections [41,42,43]. This novel application follows the proof of concept developed in the state-of-the-art with commercial agonists of such TLRs in murine models of lethal challenge with SARS and IAV [44,45]. The immunomodulatory effect of HBcAg offers potential usefulness to engineer therapies for post-exposure prophylaxis and early therapy of dengue [46] and hepatocellular carcinoma [47]. Initial results in the oncological field using HBcAg have been encouraging [48]. Long-term stimulation of innate immune responses by certain live vaccines, cytokines, Toll-like receptors and adjuvants may lead to a trained immunity, inducing heterologous protection against infection through epigenetic, transcriptional, and functional reprogramming of innate immune cells. The induction of trained immunity can be applied to reduce susceptibility to respiratory and other infectious diseases [49].

## 6. Future Prospects

Patients from a phase III clinical trial will be assessed after 2, 3, 5, and 10 years of treatment-free follow-up to understand the long-term dynamics of the virus in connection with biochemical, serological, and histological variables. Initial reports after 2 and 3 years follow-up [50,51] have shown the limited progression of liver fibrosis in CHB patients treated with Nasvac. Long-term HBsAg to anti-HBsAg seroconversion and the progression of liver fibrosis will be the major variables of the 5 to 10 years follow-up.

Considering the role of the i.n. route and its safety, a desirable target for Nasvac development is to obtain a product for i.n. administration only. The inclusion of mucoadhesive and other immunostimulants is a novel approach. Nasvac formulated with or without adjuvants will be studied in non-human primates to explore the optimal i.n. formulation. The nasal administration of Nasvac in combination with NUCs or after NUCs cessation will be evaluated in a future clinical trial. The efficacy of Nasvac vaccination associated with the immune reactivation post-NUCs may open the door to a new therapeutic scenario for a safer and more effective discontinuation of antiviral treatment.

Preclinical studies of new formulations, as well as prime-boosting approaches priming with Nasvac and boosting with live attenuated vaccine formulations are ongoing. Prime boosting approaches have been designed to further promote the immunogenicity of the vaccine in terms of specific CD8^+^ T-cell immune responses. Interesting vectors have been developed with these capabilities, as this is the case of MVA (TherVacB, TUM) and adenoviral vectors (TG1050, Transgene), which may boost the CD8^+^ T-cell response primed by Nasvac, opening the door to future combinations.

New technological developments and basic science studies have shown that the lyophilized Nasvac formulation is thermo-stable (J.C. Aguilar, personal communication), facilitating future clinical developments in remote areas or countries with economic limitations for cold chains. The mRNA vaccine technology—still under preclinical development—in the field of CHB immunotherapy may benefit from HBcAg’s capacity to store and protect the adsorbed RNA. HBcAg may be engineered in the future to encapsulate mRNA vaccines and RNA products for multiple applications as a carrier or drug delivery system.

## 7. Conclusions

The mechanism of action and the scientific rationale of the therapeutic vaccine Nasvac have been under deep scrutiny in the past two decades in multiple laboratories, trials, and animal study models. In the last century, the molecular mechanisms of the innate immune system were practically unknown. Some compounds were considered as interferon inducers; however, the pathways mediating such an effect were in the shadows. With the recent understanding of the innate immunity mechanisms, we can optimize the use of this product for CHB treatment and propose new indications and developments in the field of chronic hepatitis B and other infectious diseases. 

IFNα-based therapies have been the battle horse for decades in the immunotherapy of CHB patients. The capacity to induce adaptive, as well as innate immunity-mediated antiviral effects justifies the sustained control of the virus found in CHB patients after long-term follow-up studies. The interferon-inducing effect elicited after repeated Nasvac administration may become an adjuvant of HBV-specific adaptive immunity as well. The understanding of the action mechanism is being used to further potentiate the pharmacological effect of Nasvac in rationally designed clinical trials. Considering the role of the immunization route in the recirculation pattern of T-cells and the immunology behind the use and discontinuation of NUCs antivirals, we consider that multiple opportunities of clinical development are still open in the field of therapeutic vaccination. 

## Figures and Tables

**Figure 1 vaccines-10-02087-f001:**
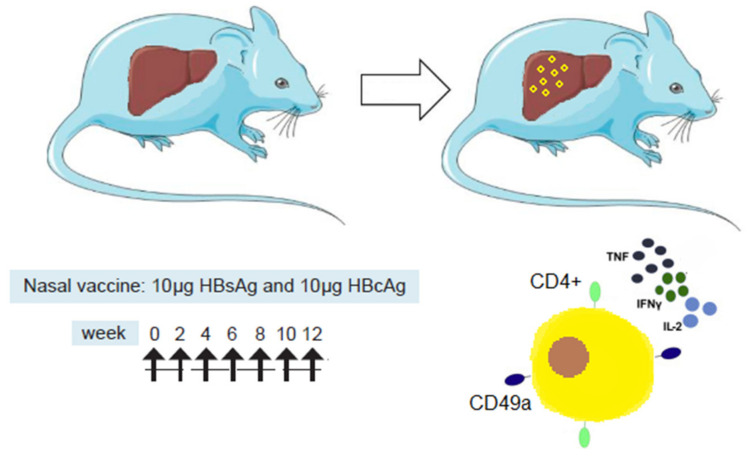
The i.n. route has the unique capacity to induce cellular immune responses, specifically CD4^+^ T-cells in the spleen and in the liver of HBV-carrier mice. Multifunctional CD4^+^ T-cells, secreting IFN-γ, IL-2, and TNF-α, are detected in the liver only after i.n. immunization. High frequencies of CD4^+^ T-cells in the liver expressing the integrin CD49a, considered a mucosal imprint, suggest a relevant role of the nasal route in the homing process.

**Table 1 vaccines-10-02087-t001:** Summary of HeberNasvac immunomodulatory and antiviral effects at the cellular and molecular levels resulting from studies in clinical trials, preclinical models, and in vitro models.

Ref.	Nasvac Study (Main Mechanisms of Action)	Pharmacological Results Unveiling the Action Mechanisms
[4]	Study of dendritic cells pulsed with both hepatitis B surface and core antigens (in vitro activation of APCs and T-cell response).	High levels of anti-HBs, HBsAg-specific and HBcAg-specific T-cells, and CTLs detected in the spleen and liver of HBV TM immunized with HBsAg-/HBcAg-pulsed DCs compared with those immunized with other vaccine formulations (*p* < 0.05). HBsAg-/HBcAg-pulsed human blood DCs also induced HBsAg- and HBcAg-specific proliferation of autologous T-cells from CHB patients.
[5]	In vitro stimulation effect of Nasvac on B- and T-cells from chronic hepatitis B patients and healthy donors (in vitro activation of B- and T-cells).	B-cell activation markers increase in CHB patients after Nasvac in vitro stimulation. B-cell activation, but not exhaustion, is induced in cells from CHB patients after Nasvac in vitro stimulation. An increase in CD25 on CD4^+^ and CD8^+^ T-cells and CD69 on CD8^+^ T-cells was observed after in vitro Nasvac stimulation. Nasvac-activated B-cells can in turn stimulate T-cells. B-cells stimulated by Nasvac may become an important APC during CHB immunotherapy.
[7,8]	Phase I/II and phase III clinical trials (in vitro APC activation and cytokine secretion pattern).	PBMCs and HBsAg-/HBcAg-pulsed DCs from HBsAg-/HBcAg-vaccinated CHB patients produced significantly higher levels of various cytokines (interleukin 1β (IL-1β), IL-6, IL-8, IL-12, and tumor necrosis factor α (TNF-α)) than those from control unvaccinated CHB patients (*p* < 0.05) after stimulation with HBsAg/HBcAg in vitro. Associated with HBV suppression in 50% of CHB patients. Generalized and transient increase of ALT at Week 12 associated with HBV control.
[12]	Study of i.n. and s.c. routes of immunization in HBV-carrier mice following a schedule of 7 doses (multifunctional T-cell response in the liver and spleen after i.n. immunization),	The i.n. route was the most efficacious at inducing cellular immune responses, in particular CD4^+^ T-cells in spleen and in the liver. Multifunctional CD4^+^ T-cells, secreting IFN-γ, IL-2, and TNF-α, were detected in the liver only after i.n. immunization. High frequencies of CD4^+^ T-cells expressing the integrin CD49a in the liver suggest a role of the nasal route in the homing process (Figure 1).
[13,14]	Study of different routes of administration, schedules of immunization, and antigen doses in HBsAg-positive transgenic mice (Ab, CD8 T-cell, and Th1 pattern of response in HBsAg tg and naïve mice).	Anti-HBsAg- and HBsAg-specific CD8+ T-cells in a tg mice model (expressing high levels of the homologous antigen in the blood, liver, and multiple organs) are induced by combining i.n. and s.c. immunization compared to the s.c. route alone. The dynamics of immune response induced in hepatitis B surface-antigen-transgenic mice immunized with Nasvac evidenced that a multiple dose schedule is a prerequisite to overcome immunotolerance in HBV-carrier mice’s Th1 pattern of response induced after i.n. administration.
[15]	HBcAg effect on HepaRG gene expression of TLR, MyD88/TRIF pathways, MHC class I/II, B7.1/2 proteins, IFNs, and cytokines and antiviral effect in HBV-infected HepaRG cells (HBcAg multi-TLR agonist effect, increasing presentation molecules, IFNs, and cytokines).	Multi-TLR agonist effect in TLR2, TLR3, TLR7, TLR8, and TLR9 cells leading to an increased stimulation of adaptors MyD88; TRIF, NF-kb, and IRF3 and molecules involved in antigen presentation (HLA class I, HLA class II, B7.1; B7.2) as well as IFNα, IFNβ, and other cytokines. Innate immune stimulation triggered a strong antiviral effect in infected HepaRG cells (similar in intensity to Entecavir).
[16]	Confirmatory clinical trial: i.n. and s.l. administration of Nasvac in >60-year-olds suspicion of SARS-CoV-2 (multi-agonist TLR expression and HLA molecules in PBMCs).	Nasvac stimulates local (TLR3, TLR,7 and TLR8 in oropharyngeal cavity) and systemic markers of innate immunity (HLA class II expression in monocytes and lymphocytes, new indication for potential use in SARS-CoV-2 post-exposure prophylaxis).

DCs: dendritic cells; PBMCs: peripheral blood mononuclear cells; CHB: chronic hepatitis B; TLR: Toll-like receptors; i.n.: intranasal; s.c.: subcutaneous; s.l.: sublingual.

## Data Availability

Not applicable.

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
