# Peer review of "Action Mechanisms and Scientific Rationale of Using Nasal Vaccine (HeberNasvac) for the Treatment of Chronic Hepatitis B"

_vaccines, 2022, doi:10.3390/vaccines10122087_

Round 1
Reviewer 1 Report
The aim of this manuscript is to discuss the action mechanism of HeberNasvac, considering the immunological features of the products and their antigens.
This manuscript shows rich content, providing a deep insight for some works: the study is within the journal’s scope, and I found it to be well-written and easy to follow, providing sufficient information. Even if the manuscript is properly organized, with a densely organized structure and based on well-synthetized evidence, there are some suggestions necessary to make the article complete and fully readable. For these reasons, the manuscript requires major changes.
Please find below an enumerated list of comments on my review of the manuscript:
INTRODUCTION:
LINE 38: Please, remove “At present” and use “Currently”, if possible.
LINE 31: Hepatitis B Virus (HBV) is a non – cytopathic, hepatotropic virus, able to induce a persistent infection, which might lead to acute and chronic diseases (see, for reference: Iannacone, M., & Guidotti, L. G. (2022). Immunobiology and pathogenesis of hepatitis B virus infection. Nature Reviews Immunology, 22(1), 19-32). HBV genome consists of a small DNA genome, which is a partially double-stranded and relaxed circular DNA (rcDNA) molecule, by creating an asymmetrical genomic structure, with a minus strand covering the whole genome and an incomplete plus strand with variable 3’ ends. In this introductive section, there is a lack of recent molecular evidence on HBV genome: the authors should highlight the genetic and molecular features of this virus, before describing the available therapeutic solutions, currently applied in the management of this infection.
LINE 44: There is no mention, in this manuscript, about the long – term immune protection, against HBV, provided by current therapeutic solutions. In this perspective, the immunogenicity elicited by HBV vaccines, has been established by several studies, conducted on children, adolescents, and adults. Furthermore, several studies and long-term follow-up studies have proved the decline of the HBsAg Ab titers, following the immunization (see, for reference: Mastrodomenico M, Muselli M, Provvidenti L, Scatigna M, Bianchi S, Fabiani L. Long-term immune protection against HBV: associated factors and determinants. Hum Vaccin Immunother. 2021 Jul 3;17(7):2268-2272. doi: 10.1080/21645515.2020.1852869. Epub 2021 Jan 31. PMID: 33522392; PMCID: PMC8189074). As the main goal of this manuscript is to evaluate the action mechanism and efficiency of this nasal vaccine, the authors should also discuss in this introductive section the importance of the persistence of long-term protection against HBV, as suggested by current evidence on this topic.
LINE 56: The authors should reformulate this sentence, in a more fluent way. Specifically: Due to its stability, this product doesn’t require preservatives or stabilizers.
LINE 83: The increase in ALT levels, in nearly all patients, temporarily associated to a generalized reduction of the HBV DNA, after the IN immunization, is suggestive of a vaccine – derived exacerbation of transaminases, developed by these patients, due to the activation of the immune response and the recognition or elimination of infected hepatocytes. The authors should rephrase this sentence, in order to improve the accuracy of the idea.
Furthermore, there is a specific and detailed explanation for the evidence included in this study: this is particularly significant, since the manuscript relies on a multitude of scientific evidence, to derive its conclusions. Scientific evidence, included in this study, are reliable and adequately discussed.
The conclusion of this manuscript is perfectly in line with the main purpose of the paper: the authors have designed and conducted the review properly. In this perspective, conclusions are well written and present an adequate balance between the description of previous findings and the results presented by the authors.
Finally, this manuscript also shows a basic structure, properly divided and looks like very informative on this topic. Furthermore, figures and tables are complete, organized in an organic manner and easy to read.
In conclusion, this manuscript is densely presented and well organized, based on well-synthetized evidence. The authors were lucid in their style of writing, making it easy to read and understand the message, portrayed in the manuscript. Besides, the methodology design was appropriately implemented within the study. However, many of the topics are very concisely covered. This manuscript provided a comprehensive analysis of current knowledge in this field. Moreover, this research has futuristic importance and could be potential for future research. However, major concerns of this manuscript are with the introductive section: for these reasons, I have major comments for this section, for improvement before acceptance for publication. The article is accurate and provides relevant information on the topic and I have some major points to make, that may help to improve the quality of the current manuscript and maximize its scientific impact. I would accept this manuscript if the comments are addressed properly.
Author Response
The aim of this manuscript is to discuss the action mechanism of HeberNasvac, considering the immunological features of the products and their antigens.
This manuscript shows rich content, providing a deep insight for some works: the study is within the journal’s scope, and I found it to be well-written and easy to follow, providing sufficient information. Even if the manuscript is properly organized, with a densely organized structure and based on well-synthetized evidence, there are some suggestions necessary to make the article complete and fully readable. For these reasons, the manuscript requires major changes.
Please find below an enumerated list of comments on my review of the manuscript:
INTRODUCTION:
LINE 38: Please, remove “At present” and use “Currently”, if possible.
We agree, in fact the expression “At present” has been removed to go straight to the point.
LINE 31: Hepatitis B Virus (HBV) is a non – cytopathic, hepatotropic virus, able to induce a persistent infection, which might lead to acute and chronic diseases (see, for reference: Iannacone, M., & Guidotti, L. G. (2022). Immunobiology and pathogenesis of hepatitis B virus infection. Nature Reviews Immunology, 22(1), 19-32). HBV genome consists of a small DNA genome, which is a partially double-stranded and relaxed circular DNA (rcDNA) molecule, by creating an asymmetrical genomic structure, with a minus strand covering the whole genome and an incomplete plus strand with variable 3’ ends. In this introductive section, there is a lack of recent molecular evidence on HBV genome: the authors should highlight the genetic and molecular features of this virus, before describing the available therapeutic solutions, currently applied in the management of this infection.
We agree, the HBV genomic description was inserted in the new version.
LINE 44: There is no mention, in this manuscript, about the long – term immune protection, against HBV, provided by current therapeutic solutions. In this perspective, the immunogenicity elicited by HBV vaccines, has been established by several studies, conducted on children, adolescents, and adults. Furthermore, several studies and long-term follow-up studies have proved the decline of the HBsAg Ab titers, following the immunization (see, for reference: Mastrodomenico M, Muselli M, Provvidenti L, Scatigna M, Bianchi S, Fabiani L. Long-term immune protection against HBV: associated factors and determinants. Hum Vaccin Immunother. 2021 Jul 3;17(7):2268-2272. doi: 10.1080/21645515.2020.1852869. Epub 2021 Jan 31. PMID: 33522392; PMCID: PMC8189074). As the main goal of this manuscript is to evaluate the action mechanism and efficiency of this nasal vaccine, the authors should also discuss in this introductive section the importance of the persistence of long-term protection against HBV, as suggested by current evidence on this topic.
We agree: The relevance/rationale of the long term follow-up efficacy will be inserted in the Introduction and give the relevance to the topic in agreement with reviewer suggestion before the start of chapter 3 where the present situation of the product is referred. In addition, a new chapter will be inserted, named “Future prospects” before Conclusions, as it was a common claim from reviewers and we have advanced in this sense. We’ll include the currently published follow-up data, as well as a new improvement in the formulation, and other studies trying to develop an IN-only product (Yoshida et al.). In this chapter we are inserting two references recently appeared in Pathogens and Vaccines that describes the 2 and 3 years follow-up of Nasvac treated patients, in summary. Subsequently we will mention that we are following the patients after 5 and 10 years to have a better picture of HBsAg reduction, liver disease progression and other serological, biochemical and virological markers.
References:
Akbar, S.M.F.; Al Mahtab, M.; Aguilar, J.C.; Yoshida, O.; Penton, E.; Gerardo, G.N.; Hiasa, Y. Sustained Antiviral and Liver Protection by a Nasal Therapeutic Vaccine (NASVAC, Containing Both HBsAg and HBcAg) in Patients with Chronic Hepatitis B: 2-Year Follow-Up of Phase III Clinical Trial. Pathogens, 2021, 10(11):1440. DOI: 10.3390/pathogens10111440.
Akbar, S.M.F.; Al Mahtab, M.; Aguilar, J.C.; Yoshida, O.; Khan, S.; Penton, E.; Gerardo, G.N.; Hiasa, Y. The Safety and Efficacy of a Therapeutic Vaccine for Chronic Hepatitis B: A Follow-Up Study of Phase III Clinical Trial. Vaccines (Basel), 2021, 10(1), 45. DOI: 10.3390/vaccines10010045.
Yoshida, O.; Akbar, S.M.F.; Imai, Y.; Sanada, T.; Tsukiyama-Kohara, K.; Miyazaki ,T.; Kamishita, T.; Miyake, T.; Tokumoto, Y.; Hikita, H.; Tsuge, M.; Shimizu, M.; Al Mahtab, M.; Aguilar, J.C.; Guillen, G.; Kohara, M.; Hiasa, Y. Intranasal therapeutic vaccine containing HBsAg and HBcAg for patients with chronic hepatitis B; 18 months follow-up results of phase IIa clinical study. Hepatol Res. 2022 Nov 18. doi: 10.1111/hepr.13851.
LINE 56: The authors should reformulate this sentence, in a more fluent way. Specifically: Due to its stability, this product doesn’t require preservatives or stabilizers.
We agree. Suggestion accepted.
LINE 83: The increase in ALT levels, in nearly all patients, temporarily associated to a generalized reduction of the HBV DNA, after the IN immunization, is suggestive of a vaccine – derived exacerbation of transaminases, developed by these patients, due to the activation of the immune response and the recognition or elimination of infected hepatocytes. The authors should rephrase this sentence, in order to improve the accuracy of the idea.
We have rephrased the sentence.
Furthermore, there is a specific and detailed explanation for the evidence included in this study: this is particularly significant, since the manuscript relies on a multitude of scientific evidence, to derive its conclusions. Scientific evidence, included in this study, are reliable and adequately discussed.
The conclusion of this manuscript is perfectly in line with the main purpose of the paper: the authors have designed and conducted the review properly. In this perspective, conclusions are well written and present an adequate balance between the description of previous findings and the results presented by the authors.
Finally, this manuscript also shows a basic structure, properly divided and looks like very informative on this topic. Furthermore, figures and tables are complete, organized in an organic manner and easy to read.
In conclusion, this manuscript is densely presented and well organized, based on well-synthetized evidence. The authors were lucid in their style of writing, making it easy to read and understand the message, portrayed in the manuscript. Besides, the methodology design was appropriately implemented within the study. However, many of the topics are very concisely covered. This manuscript provided a comprehensive analysis of current knowledge in this field. Moreover, this research has futuristic importance and could be potential for future research. However, major concerns of this manuscript are with the introductive section: for these reasons, I have major comments for this section, for improvement before acceptance for publication. The article is accurate and provides relevant information on the topic and I have some major points to make, that may help to improve the quality of the current manuscript and maximize its scientific impact. I would accept this manuscript if the comments are addressed properly.
We really appreciate the warm words of our reviewer as well as his effort to provide or suggest solutions. His revision has significantly helped to rapidly process the article. We would like to express our gratitude. All our answers are all in line with the received suggestions.
The authors
Reviewer 2 Report
Dear colleague,
Many thanks for reviewing the current literature on and bringing the aspects of HeberNavac to the attention of scientific community.
This is a well-written manuscript with a scientific premise.
Author Response
Dear colleague,
Many thanks for reviewing the current literature on and bringing the aspects of HeberNavac to the attention of scientific community.
This is a well-written manuscript with a scientific premise.
Answer/
We highly appreciate reviewers opinion. Some changes were inserted to accomplish suggestions from other reviewers, and to improve quality aspects of the manuscript requested by the editors. In case you find interesting to insert or delete information, we will be happy to modify the document accordingly, however, we consider the resulting article doesn’t contradict the initial version, in contrast the quality of the text and discussion has increased.
Please, receive our gratitude
The authors
Reviewer 3 Report
1. Recently therapeutic vaccine is in the spotlight because of SAS-cov-2 mRNA vaccine emergence. This review introduced NASVAC as a therapeutic vaccine against CHB which is interesting to correlated with hot mRNA therapeutic vaccine.
2. More information should be collected such as NASVAC is also named ABX203,
3. There are two clinical trials in https://clinicaltrials.gov/, as a review, both of these two trials should be discussed. But the authors only mentioned one of it.
4. In the clinicaltrials website, NCT01374308 was opened in 2011, NCT02249988 was first posted in 2014. The sponsors of two studies are different, also 10 year passed. The author should update the NASVAC status, and further discuss the reason of NASVAC hasn’t latest information. Of course in line 276-281, the authors talked a novel trial, more details including reasonale of this study should be discussed.
5. Summary of HeberNasvac immunodulatory and antiviral effects at the cellular and molecular levels.
The summary in table.1 should be classified by mechanism of action, such as humoral immunity, cytokines and TLRs were stimulated after vaccination.
6. I noticed NASVAC was administered to 160 CHB patients in several doses, but other therapeutic vaccines usually are less than 3 doses. As we know, vaccines could ignite B memory and T memory immunity, if NASVAC also stimulated enough memory immunity should be discussed.
7. This product was administered by nasal and subcutaneous routes. The analysis and introduction of different immunity response by these two routes is required.
8. HBcAg showed multi-toll-like receptors (multi-TLR) agonist effect, actually several TLRs agonists including TLR7, TLR8 and TLR9 were developed to against HBV treatment by many company. Here the authors should discuss the relationship.
9. The clinical development strategies of NUC and NASVAC are well discussed by authors. However, the details are mess and duplicated. From line 167 to line 281, each section is too long and the content should be reorganized and added subheading. For example, first the authors could introduce disadvantage of NUC, then talk combination NUC and at last mention NASVAC as a new opportunity for immunotherapy.
10. Other minor issue, too many numbers 4 appeared at the beginning of chapter.
11. Therapeutic vaccine of other competitors against HBV such as GenHevac B, DV601, (Theravax) and so on should be discussed.
12. In the conclusion, the advantage/disadvantage and development in the future of NASVAC should be discussed.
Author Response
Dear reviewer,
We start expressing our gratitude for the critical revision. In general, we have taken into account all your suggestions. In all cases we have answered with our point of view, in most cases we have introduced the modifications in the text and in some cases, when the information was there -but not clearly expressed, we have modified the wording to get a more comprehensive manuscript. In summary, we appreciate all comments and consider it has been a great support to improve the quality of the manuscript. We hope all aspects were considered and clarified; otherwise we will be happy to further amend our document. Find below a point by point answer:
- Recently therapeutic vaccine is in the spotlight because of SAS-cov-2 mRNA vaccine emergence. This review introduced NASVAC as a therapeutic vaccine against CHB which is interesting to correlate with hot mRNA therapeutic vaccine.
We agree with the reviewer that the mRNA vaccine technology opens new opportunities; however, there is limited information on the use of mRNA vaccination for treatment of CHB. In a recent editorial article in the field of Hep B, authors highlight DNA vaccination, contrasting with the field of mRNA-based vaccines, “limited to a few in vitro experiments in this area”, concluding that “Further studies are needed to clarify the prospects of nucleic acid vaccines for HBV cure”. [Tsounis, E.P.; Mouzaki, A.; Triantos, C. Nucleic acid vaccines: A taboo broken and prospect for a hepatitis B virus cure. World J Gastroenterol, 2021, 27(41), 7005-7013. DOI: 10.3748/wjg.v27.i41.7005.].
In the case of mRNA vaccination in CHB patients, as well as in the case of NASVAC –that contains an important amount of RNA associated to HBcAg-, the effect on innate immunity stimulation needs consideration. In a recently published article, studying Covid-19 vaccination in CHB patients it was observed that the number of patients whose HBsAg levels was reduced by >50% per year was prominent after the initiation of the vaccination program. Authors suggested that mRNA COVID-19 vaccines may have been involved in HBsAg reduction in patients with chronic hepatitis B. [Osawa, Y.; Ohtake, T.; Suto, D.; Akita, T.; Yamada, H.; Kohgo, Y.; Murata, K. Cases of Rapid Hepatitis B Surface Antigen Reduction after COVID-19 Vaccination. Intern Med, 2022, Oct 19 Online ahead of print. DOI: 10.2169/internalmedicine.0842-22.].
Considering previously mentioned observations we agree with the reviewer that we may find points of contact between Nasvac and mRNA vaccines. Moreover, we are planning to use HBcAg as a carrier for mRNA vaccines to improve the thermo-resistance, biodistribution and reduction in the number of immunizations in future. We have inserted a new chapter “Future Prospects” mentioning the limited development of mRNA vaccines in the field of CHB immunotherapy and our interest to use the HBcAg as a potential carrier for mRNA vaccines in future developments.
- More information should be collected such as NASVAC is also named ABX203,
The information of NASVAC when coded as ABX203 (given in combination with NUCs) is described in the manuscript. See Chapter entitled: Nasvac vaccination and the cessation of NUCs. The product used in both settings was the same in terms of composition, dose, route and quality specifications. The only difference was the combination with antivirals in the study of ABX203 [Reference 10, below]. In the ABX203 study, the decision to reintroduce patients back to NUC treatment after viral rebound prevented subsequent assessment of the patients (it was discussed with the team of doctors and hepatologist’s).
We are clarifying that ABX203 and NASVAC vaccine composition, dose, route and quality specifications were the same in both studies next to the description of the study. We will also insert the NCT number to the ABX trial as explained in the answer to your query No. 3.
Reference 10: Wedemeyer, H.; Hui, A.J.; Sukeepaisarnjaroen, W.; Tangkijvanich, P.; Su, W.W.; Nieto, G.E.; Gineste, P.; Nitcheu, J.; Crabé, S.; Stepien, S.; Cornberg, M.; Trépo C. Therapeutic vaccination of chronic hepatitis B patients with ABX203 (NASVAC) to prevent relapse after stopping NUCs: contrasting timing rebound between tenofovir and entecavir. J Hepatol 2017, 66: S101. DOI: 10.1016/S0168-8278(17)30463-4.
- There are two clinical trials in https://clinicaltrials.gov/, as a review, both of these two trials should be discussed. But the authors only mentioned one of it.
Both clinical trials are discussed in the manuscript:
NCT01374308: it was discussed in Chapter 3.1 Association of the i.n. immunization to immune activation and HBV clearance.
NCT02249988: it was discussed in the “new” chapter 5, Nasvac vaccination and the cessation of NUCs, (previously number 4).
To further clarify this, we will insert the NCT reference corresponding to the study named: “(Clinical Efficacy of ABX203 Therapeutic Vaccine in HBeAg Negative Patients With Chronic Hepatitis B)”.
- In the clinicaltrials website, NCT01374308 was opened in 2011; NCT02249988 was first posted in 2014. The sponsors of two studies are different, also 10 year passed. The author should update the NASVAC status, and further discuss the reason of NASVAC hasn’t latest information. Of course in line 276-281, the authors talked a novel trial, more details including reasonale of this study should be discussed.
Early in the manuscript, -during the introduction of the product at chapter 2 [2. Nasvac: a novel vaccine for chronic hepatitis B treatment], it is described the current status of Nasvac:
…Nasvac has completed preclinical and clinical steps in Cuba and Bangladesh as part of an international effort contributing scientists from Cuba, Bangladesh, Japan, Australia, France and Germany [3-8]. The regulatory process is advancing with different speed in several countries, where the product has been studied as a monotherapy and also as a combined treatment with the NUCs [9, 10]. At present, it has been registered for monotherapy of CHB in Cuba [3] and different studies are optimizing the use of this product as a combined therapy, however, the combination of immunomodulatory drugs and antivirals has proven challenging and deserve a further immunological understanding [2,10,11]
We agree with the reviewer that we didn’t mentioned the two articles published in recent years compiling the follow-up data of Nasvac. Following the suggestions of the reviewer, we will insert the two references in the manuscript before Conclusions in a chapter that will be named: Future prospects. This information will be allocated in this chapter considering that one of our future prospects is to follow the patients during 10 years and the publication of the results correspond to 2 and 3 years follow-up.
Akbar, S.M.F.; Al Mahtab, M.; Aguilar, J.C.; Yoshida, O.; Penton, E.; Gerardo, G.N.; Hiasa, Y. Sustained Antiviral and Liver Protection by a Nasal Therapeutic Vaccine (NASVAC, Containing Both HBsAg and HBcAg) in Patients with Chronic Hepatitis B: 2-Year Follow-Up of Phase III Clinical Trial. Pathogens, 2021, 10(11):1440. DOI: 10.3390/pathogens10111440.
Akbar, S.M.F.; Al Mahtab, M.; Aguilar, J.C.; Yoshida, O.; Khan, S.; Penton, E.; Gerardo, G.N.; Hiasa, Y. The Safety and Efficacy of a Therapeutic Vaccine for Chronic Hepatitis B: A Follow-Up Study of Phase III Clinical Trial. Vaccines (Basel), 2021, 10(1), 45. DOI: 10.3390/vaccines10010045.
We have described that Nasvac vaccine candidate is following a novel approach, where it is applied to patients just after NUC cessation in order to bring together the natural effect of immune activation and viral control after the HBV DNA rebound together with the vaccine induced immune response. As it is supposed to provide preliminary results by next years, it was mentioned also include in the “new” chapter “Future prospects”. In addition, we will inform about the new formulations being tested in clinical trials by the Ehime University in Japan in recent years as the article has just been published, the new ref was inserted. This study will be mentioned in future prospect as well because this is a new formulation development comprising the mucoadhesive carboxy-vinyl polymer (CVP) and studies in non-human primates comparing all formulations with potentialities to become an i.n. only vaccine, in an attempt to explore whether the s.c. administration might be circumvented.
- Summary of HeberNasvac immunodulatory and antiviral effects at the cellular and molecular levels.
The summary in table.1 should be classified by mechanism of action, such as humoral immunity, cytokines and TLRs were stimulated after vaccination.
The suggested classification was added in the table.
- I noticed NASVAC was administered to 160 CHB patients in several doses, but other therapeutic vaccines usually are less than 3 doses. As we know, vaccines could ignite B memory and T memory immunity, if NASVAC also stimulated enough memory immunity should be discussed.
We agree with the reviewer that a large number of articles have been written, characterizing the immune response of vaccine candidates. An ideal vaccine should have the minimum number of doses that may overcome a well-established tolerance in a safe and effective way [safe control of HBV, serological responses and eventually the elimination of HBsAg, preventing or reducing liver damage.
Due to the insufficient effect or the absence of efficacy, the therapeutic vaccination with most of these formulations has been disappointing during their clinical development. The new reality of treating patients with NUCs, an important achievement for patients, further complicated therapeutic vaccine development as many of the studies were unable to stop NUCs due to ethical reasons and the fear of uncontrolled activation of immune response. We explain the immunological scenario in the manuscript in a well-organized section with specific subheadings as requested by the reviewer.
Our initial studies in naïve mice evidenced the capacity of the formulation to induce an adaptive and long lasting immune response [ref 35, commented in the table], the subsequent analysis of the immune response in the model of immunotolerance evidenced that the maximum Ab and CD8+ T cell response was induced after the fifth dose, and that the vaccine could be safely administered for a large number of administrations [ref 13 and 14], these results were already summarized in the table, the text linked to these references will be modified to ensure a better comprehension.
More importantly, and also considered in the article, the ALT increases during phase III clinical trial using Nasvac as monotherapy were consistent in the fact that most patients increased their ALT levels in a transient flare between 100 to 200 ALT units/L after the fifth IN administration with NASVAC (after the IN cycle, dose No 5). Coming back to the bench, we realized that the IN route was critical to detect specific and multifunctional anti HBV CD4+ T cells in the liver of mice expressing the HBV DNA and other serological markers after a 7 dose immunization schedule, study conducted in collaboration with the developers of the novel model of specific HBV immunotolerance at Pasteur Institute. The study detected a high frequency of mucosal imprinting in the intrahepatic HBV specific multifunctional CD4+ T cells as well.
The relevance of the immunological data should be considered when the formulation is capable of inducing the immune response in the target organ [see Fig 1], that’s why we highlighted these results in our review article. In our analysis of product potentialities of use, we found that the only finite treatment in the market used 48 weekly administrations. Thus, the product was competitive enough in terms of immunization schedule, costs, safety and several variables of efficacy, although further studies are required to expand the clinical experience with the product and to assess IN-only administration schedules.
In summary, although some references and results are already inserted in the manuscript, to further clarify the relevance of the immunization schedule and the memory T cell response, we propose to stress the above mentioned ideas both at the table and summarized next to the new references inserted with the long term follow-up clinical data. We will highlight the relevance of the immunization schedule with five or more doses as well as the late response and long lasting responses evidenced in clinical trials. Also, it will be recognized that further optimizations in terms of dose number and immunization route may be achieved in future; specifically in terms of CD8+ T cell responses.
- This product was administered by nasal and subcutaneous routes. The analysis and introduction of different immunity response by these two routes is required.
We are inserting more details in the table associated to the references 13 and 14 where the effect of the s.c or i.n./s.c. simultaneous immunizations was compiled. We have an additional papers with information at this regard that was not inserted (see ref below), however, with the modification of the text in the table, we considered the point highlighted by the reviewer is adequately supported.
Lobaina, Y.; Trujillo, H.; García, D.; Gambe, A.; Chacon, Y.; Blanco, A.; Aguilar, J.C. The effect of the parenteral route of administration on the immune response to simultaneous nasal and parenteral immunizations using a new HBV therapeutic vaccine candidate. Viral Immunol 2010, 23(5):521-9. DOI: 10.1089/vim.2010.0024.
- HBcAg showed multi-toll-like receptors (multi-TLR) agonist effect, actually several TLRs agonists including TLR7, TLR8 and TLR9 were developed to against HBV treatment by many company. Here the authors should discuss the relationship.
As per reviewer suggestion we will highlight the relevance of using a multi-TLR agonist in our therapeutic vaccine. We will insert this text in the manuscript in chapter: “On the immunomodulatory properties of Nasvac antigens” , giving a prominent space to the field of CHB, main focus of the article and linking the effect to the state of the art that justifies the use of TLRs in CHB treatment as commented by the reviewer.
Inserted text:
The multi-TLR agonist effect is relevant in CHB immunotherapy considering the strategies of HBV to evade the innate immune response and induce immunosuppression. PBMCs isolated from CHB patients have decreased expression of TRIF, attenuating TLR3 and TLR4 signaling subsequent to HBV infection. This has been considered as one of the reasons why HBV infection is stable in CHB patients as recently reviewed by Xu and colleagues [40]. TLR2 is involved in the recognition of HBV during the early infection process and initiates innate immune signaling in primary human hepatocytes. Several TLRs, including TLR3, TLR4, TLR7, and TLR9, are downregulated in CHB patients. In vitro studies using the HBcAg evidenced a strong reduction in HBV replication in cultured HepaRG cells [36]. This pharmacological effect of HBcAg on the innate immune stimulation -as IFN inducer- may also mediate a potential antiviral effect in addition to the adaptive immune response in vivo.
A new reference is inserted in support of these statements, a relevant review in this area:
- Xu, C.; Chen, J.; Chen, X. Host innate immunity against hepatitis viruses and viral immune evasion. Front Microbiol, 2021 12, 740464. DOI: 10.3389/fmicb.2021.740464
- The clinical development strategies of NUC and NASVAC are well discussed by authors. However, the details are mess and duplicated. From line 167 to line 281, each section is too long and the content should be reorganized and added subheading. For example, first the authors could introduce disadvantage of NUC, then talk combination NUC and at last mention NASVAC as a new opportunity for immunotherapy.
Done! I have, in addition, eliminated information that was too long, also introduced the subheadings, 100% in agreement, thanks!
- Other minor issue, too many numbers 4 appeared at the beginning of chapter.
It was corrected in the new version of the manuscript.
- Therapeutic vaccine of other competitors against HBV such as GenHevac B, DV601, (Theravax) and so on should be discussed.
We are already mentioning the competitors, when we mention that they have been essentially stopped their program as a result of clinical trials that have been conducted under antivirals and with a very strict rule for their reintroduction under NUC therapy. References to these competitors are [16-18], after mentioning them, we focused in potential immunological rationale of combining such therapeutic vaccines with NUCs.
In support of reviewer suggestion, in the chapter “Future prospects”, we will mention the potential use of Nasvac as a priming option for TherVaxB, an ongoing project that may further potentiate the CD8+ T cell response by the booster with the MVA vaccine. This is an already existing assessment in preclinical studies that deserve a future clinical evaluation.
- In the conclusion, the advantage/disadvantage and development in the future of NASVAC should be discussed.
We have inserted the developments of Nasvac in the chapter “Future prospects”, introducing the issue of advantages / disadvantages of HeberNasvac, and reinforcing the Conclusions subsequently in line with reviewer observations.
We highly appreciate reviewer considerations and suggestions as we understand the article was very much enriched, in summary we agree with all recommendations and hope we cover the expectations. In case an additional amendment is required we will be happy to work on it asap.
The authors
Round 2
Reviewer 1 Report
well done
Reviewer 3 Report
The authors appropriately answer all the questions and the revised version was great improved.